# Self-Efficacy, Perceived Stress, and Individual Adjustment among College-Attending Emerging Adults

Rebecca C. Madson [1,*], Paula B. Perrone [1], Sara E. Goldstein [2] and Chih-Yuan Steven Lee [1]

1   Department of Family Science and Human Development, Montclair State University,
    Montclair, NJ 07043, USA
2   Department of Human Development and Family Sciences, University of Delaware, Newark, DE 19716, USA
*   Correspondence: madsonr@montclair.edu

**Abstract:** In a large, ethnically diverse sample of college-attending emerging adults (N = 693; ages 18–29), the current study examines associations between self-efficacy and individual adjustment (academic satisfaction, depressive symptoms, subjective physical health, and loneliness), directly and indirectly through perceived stress. Moderated mediation effects by sex, ethnicity, school year, and first-generation status were also explored. Using PROCESS, results show that self-efficacy was directly related to adjustment, and indirectly related through lower stress. Sex moderated the associations between self-efficacy and stress as well as stress and depressive symptoms; the relations were stronger in women. School year moderated how stress was associated with academic satisfaction in that the negative association was not found among the fourth-year students, but in all other peers. First-generation status moderated the negative association of self-efficacy and stress, with it being greater for first-generation college students compared to their peers. In addition, self-efficacy was positively related to academic satisfaction for first-generation students, but no relation was found for other students.

**Keywords:** self-efficacy; perceived stress; individual adjustment; emerging adulthood; well-being; college students

## 1. Introduction

Emerging adulthood is a developmental period characterized by transition, change, and exploration [1]. Ranging from approximately ages 18 through 29, this can be a very exciting time, but it also can be challenging. For many college-attending emerging adults, their time at university is marked by a variety of mental health concerns including stress, anxiety, depression, and loneliness [2–5]. These issues can be compounding; mental health challenges present increased risk for academic failure and drop-out, especially during the first two years of university, and for first-generation students [6–8].

Given the psychosocial and academic challenges associated with this developmental period, especially for at-risk students, it is important to better understand factors associated with positive adjustment among college-attending emerging adults. Stress during this time is ubiquitous; however, some youth experience relatively high levels in comparison to their peers [2]. Generally speaking, high levels of stress are associated with mental health struggles such as depression [9–11]. However, stress has different impacts across individuals; other aspects of individuals' experiences can mitigate the impacts of stress. For example, in a sample of emerging adults, Lee et al. [5] found that family support mediated the relations between stress and physical health. In contrast, peer support (from friends and romantic partners) mediated the associations of stress with loneliness and depressive symptoms.

Social–cognitive factors play an important role in determining the impact of stressful experiences [12]. Self-efficacy, notably, is critical for both student persistence [13] and

psychosocial adjustment [14,15]. When one believes that they are able to accomplish a goal and achieve success, they are more likely to do so. There are direct associations between stress and wellness, as well as between stress and academic achievement [9,16,17]. It remains unknown, however, how stress may mediate associations between self-efficacy and adjustment, or whether these relations may vary based on sex, race, school year (e.g., first-year, second-year), or first-generation college status (i.e., whether the student is the first in their family to attend university). Although existing studies may focus on perceived stress and self-efficacy, they tend to focus on one facet of individual adjustment (such as life satisfaction) rather than multiple aspects [18,19].

Additionally, this study focuses on a large, ethnically diverse sample. Many studies related to self-efficacy, perceived stress, and adjustment have come from culturally homogenous groups, specific subgroups of college-attending emerging adults (such as first-year students, specific academic programs, etc.) or smaller samples. Given the diversity of college-attending emerging adults in the United States and around the world, a heterogenous sample can help illuminate the spectrum of experiences of a variety of populations.

## 1.1. Self-Efficacy

Self-efficacy is a concept that helps us understand how people persist in the face of obstacles and adversity [12]. Bandura's social cognitive theory focuses on human agency, or how people express power, make choices, and act on them [20]. Self-efficacy is an individual's belief that their agency will have certain results. Although the concept of self-efficacy is not new, it remains an important topic in understanding the experience of college-attending emerging adults. Emerging adulthood, and the pursuit of higher education, is a time marked by many changes and challenges [1]. Additionally, for many, it is the beginning of their independence from the family unit, and the start of their taking responsibility of their actions, as well as gaining awareness of their abilities [20]. Thus, self-efficacy plays a significant role in comprehending which college-attending emerging adults will succeed and thrive, and, according to Bandura, may interact with further social cognitive variables to predict adjustment as discussed further below [20].

## 1.2. Self-Efficacy and Individual Adjustment

Self-efficacy is a particularly powerful factor in terms of understanding behavior in unfamiliar experiences [12]. This is especially important for the college years, when students will come across many new challenges. Thus, it is not surprising that self-efficacy has been linked to a number of factors associated with mental health for college-attending emerging adults. For example, numerous studies have found that self-efficacy is negatively related to depression in college students [15,21–23]. Self-efficacy as a by-product of social support is also negatively correlated with depressive symptoms [24] and loneliness in college students [25]. Self-efficacy is seen as one of the factors of motivation, which is strongly and positively correlated with student persistence and academic achievement [13]. Self-efficacy is also positively related with adjustment [26–29], including psychological adjustment [30] and college academic performance [14,31–35].

## 1.3. Self-Efficacy, Perceived Stress, and Adjustment

When one believes as though they will succeed in a given task, one tends to be less stressed or bothered by that task. A number of studies have found a negative correlation between self-efficacy and perceived stress in adolescents [18], in first-year college students [36,37], and in female undergraduate students [38]. Students who report high self-efficacy may see college as a challenge, which speaks of opportunity, versus a threat, which has a negative connotation [39].

In contrast, perceived stress, without self-efficacy to counteract it, is a significant predictor of negative academic performance in students [17]. Chronic stress is associated with psychological distress in college students, characterized by anxiety, depressive symp-

toms, and loss of emotional/behavioral control [40]. Similarly, chronic and daily stress are associated with maladaptive health behaviors [16]. Thus, stress in the absence of other positive cognitive framing strategies (such as self-efficacy) can have a deleterious impact on emerging adults' adjustment [27].

*1.4. Moderator Effects*

As noted above, self-efficacy and stress do not uniformly impact people who experience them. Previous research suggests that sex, race/ethnicity, school year, and first-generational college status may all be important variables in understanding relations among self-efficacy, stress, and individual adjustments [4,39,41–44].

1.4.1. Sex

Sex is an important factor to consider with regard to understanding the implications of stress. For example, a study on stress and depressive symptoms among college students found that male students had depressive symptoms related to academic and interpersonal challenges, whereas female students' symptoms were related to social activities and intrapersonal problems [9]. Additionally, this study reported higher overall levels of stress for the females, as compared to their male peers [9]. In another study focusing on adult women versus adult men, Maciejewski et al. [45] found that women were at greater risk of suffering from depressive symptoms as the result of a stressful life event. Moreover, in research focusing on the transition to college, there is evidence that female students are more likely than their male peers to suffer extended psychological distress during their first year of college [4]. In another study related to self-efficacy, while there was not a significant difference in reported self-efficacy between sexes, female students fared better than their male counterparts in adjustment [26].

1.4.2. Race/Ethnicity

Research has found that self-efficacy, and its implications, can also vary based on cultural backgrounds. For example, individuals from cultures that value collectivism tend to score lower on measures of self-efficacy, as compared to individuals from cultures that emphasize individualism [46]. In addition, students from collectivist cultures have been shown to perceive greater amounts of stress, as compared to their peers from more individualistic backgrounds [46]. Similarly, self-efficacy can vary for racially minoritized students. To illustrate, Hermann and Betz [21] found that African American students reported higher levels of self-efficacy than their peers. In a study of mostly immigrant and minority first-year students, self-efficacy and stress were negatively correlated [36]. This study also found that self-efficacy was the greatest predictor of academic performance, as measured by GPA [36].

1.4.3. School Year

In recent years, increased attention has been paid to first-year college students and understanding the challenges they face, particularly around stress, health, and adjustment, during their transition into college [4,25,26,31,36,39,47–49]. For this population in particular, adjustment during the first year is critical, as it may influence their ability to be retained by the institution and continue with their studies. Towbes and Cohen's [40] study, consisting of students from all four years, found that first-year students scored the highest on chronic stress. Other studies have focused on students in their last year of university. These "seniors" (i.e., students in their last year of studies) have shown unique patterns of results, perhaps due to a sort of restriction of range in terms of the characteristics of students who make it to their final year of studies. For example, in a study of senior students the population did not display much variety in self-efficacy, resilience, and persistence, regardless of sex or ethnicity. This may be explained by the fact that they had progressed and made it to their final year [50].

Interestingly, most research on this topic has focused on first- or last-year college students (either transitioning into or out of college) and has typically not focused on the "middle" college years. This is unfortunate, as these years are critical for academic and occupational development and tend to be the years when students become increasingly invested in their majors and major-associated career opportunities [8,51]. In an exception, one study of first-generation students found that sophomore students performed better academically than their first-year peers, and that their academic performance was highly correlated with academic self-efficacy [35].

### 1.4.4. First-Generation

Students who are the first in their family to attend college face unique challenges. This group of students is less likely to complete a college degree than their continuing-generation peers, which could be a result of having fewer financial resources or less academic preparation [7,52]. A few studies have explored first-generation students' self-efficacy [35,52,53], but less is known about the associations between stress and adjustment for this population. Some studies have found that first-generation college students have lower academic self-efficacy [53], whereas others find that first-generation students' self-efficacy matches that of their continuing-generation peers, at least in senior students [50]. It is worth noting that for first-generation students, increasing self-efficacy has shown to be an even more important factor for academic performance than their continuing-generation peers [31].

### 1.5. Current Study

Given the issues described above, the current research examines associations between self-efficacy and individual adjustment; both directly and indirectly through perceived stress. These issues are explored in a large and ethnically/racially diverse sample of college-attending emerging adults. We also explore whether both direct and indirect associations are moderated by sex, ethnicity, school year, or first-generation college student status. To capture diverse aspects of individual adjustment during college, overall academic satisfaction, depressive symptoms, loneliness, and subjective physical health are all considered. In this study, we will examine the following hypothesis (H) and research question (RQ):

H: Perceived stress mediates the association between self-efficacy and individual adjustment.

RQ: How does the mediating association between self-efficacy, perceived stress, and individual adjustment differ based on sex, race/ethnicity, school year, and first-generation status?

## 2. Materials and Methods

### 2.1. Procedure and Participants

The sample consisted of 693 emerging adults attending a mid-sized public university in the Northeastern United States (80.2% female; $M_{age}$ = 20.03 years; $SD_{age}$ = 1.99; Range = 18–29). Regarding ethnicity, the sample was 49.6% White, 12.7% Black/African American, 26.1% Latina/o, 3.5% Asian American, 6.2% multiracial, and 1.9% reported an "other" ethnic background. Participants were volunteers; they were recruited in classrooms and through word of mouth. All data were collected in person, and all procedures were approved by the university's Institutional Review Board. After providing informed consent, participants completed a survey that included demographic questions and the following measurement instruments.

### 2.2. Measures

#### 2.2.1. Self-Efficacy

Self-efficacy was measured by the Generalized Self-Efficacy Scale [54], aimed to assess a general sense of perceived self-efficacy. Using a 4-point scale, participants were asked to respond to 10 statements that best describe their responses. Sample statements were:

"I can always manage to solve difficult problems if I try hard enough" and "I can usually handle whatever comes my way" (1 = *not at all true*, 2 = *hardly true*, 3 = *moderately true*, 4 = *exactly true*). Reliability in the current sample was good ($\alpha = 0.88$). Higher scores on the scale denote increased levels of self-efficacy.

### 2.2.2. Perceived Stress

We assessed perceived stress using the 10-item Perceived Stress Scale (PSS-10) [55]. This instrument measures the degree to which various situations a person encounters in life are experienced as stressful. Applying a 5-point scale (0 = *never* to 4 = *very often*), respondents rated how often over the prior month they felt or thought certain ways, such as being upset by something happening unexpectedly, or feeling unable to exert control over key matters in their lives. Higher scores indicated higher perceived stress levels. Reliability in the current sample was good ($\alpha = 0.82$).

### 2.2.3. Overall Academic Satisfaction

Overall academic satisfaction was assessed by asking the participants a single question, "Overall, how satisfied are you with your school performance?" (1 = *very dissatisfied*, 2 = *dissatisfied*, 3 = *satisfied*, 4 = *very satisfied*). Satisfaction with performance is conceptually linked to actual performance but provides a more direct indication of psychological adjustment by prioritizing how students feel about their performance, regardless of their actual performance [56].

### 2.2.4. Depressive Symptoms

Depressive symptoms were measured by the Center for Epidemiologic Studies-Depression scale (CES-D) [57]. Participants were asked to indicate in the past week how often they might have felt in response to each of the 20 ways listed, including "I felt sad" and "I had crying spells" (1 = *rarely or none of the time*, 2 = *some or little of the time*, 3 = *occasionally or a moderate amount of time*, 4 = *most or all of the time*). Higher scores indicate higher levels of psychological distress. Reliability in the present sample was excellent ($\alpha = 0.90$).

### 2.2.5. Loneliness

Loneliness was evaluated by the 8-item short-form of the UCLA Loneliness Scale (UCLA-8) [58]. Participants rated how often they felt the way described in each of the eight statements (1 = *never*, 2 = *rarely*, 3 = *sometimes*, 4 = *often*). Sample statements include, "I feel isolated from others" and "I lack companionship." Mean scores were calculated so that higher scores signify higher levels of loneliness. Reliability in the current sample was good ($\alpha = 0.83$).

### 2.2.6. Subjective Physical Health

Physical health was measured using a single question, to which participants responded by evaluating their health in general using a 5-point scale (1 = *poor* to 5 = *excellent*). One-item scales relying on self-ratings have long been used to assess individuals' subjective physical heath [59], and validity of the evidence has been established via strong correlations between ratings of health and immune system functioning provided by physicians, as well as mortality [60,61].

### 2.3. Analysis Strategy

PROCESS [62] was utilized to test the proposed mediator model for each of the four adjustment indicators separately (H). PROCESS is a regression-based computational tool that can test path analysis-based mediation and moderation. For continuous outcomes, PROCESS uses OLS regression to estimate unstandardized model coefficients, standard errors, *t* and *p*-values, and confidence intervals. In mediation models, PROCESS estimates direct effects (*c'*) as well as indirect effects (*ab*) through bootstrapping. In this study, the indirect effects were tested with 10,000 bootstrapped samples and a bias-corrected 95%

confidence interval (CI). The indirect effects are statistically significant when zero is not located in the CI.

PROCESS was also employed to explore conditional process models of direct and indirect effects (moderated mediation models; RQ). Specifically, we explored whether sex, race/ethnicity, school year, and first-generation status each moderated the direct and indirect associations between self-efficacy and individual adjustment. Whenever any conditional effects were identified, we followed the guidelines provided by Aiken and West [63] to probe and interpret the results. Table 1 presents the correlation matrix with means and standard deviations for the predictor/mediator variables, moderator variables, and criterion variables in the study.

**Table 1.** Means, Standard Deviations (SDs), and Intercorrelations among Study Variables (*N* = 693).

|                            | 1.       | 2.       | 3.       | 4.      | 5.        | 6.        | 7.        | 8.        | 9.        | 10. |
|----------------------------|----------|----------|----------|---------|-----------|-----------|-----------|-----------|-----------|-----|
| 1. Gender                  | —        |          |          |         |           |           |           |           |           |     |
| 2. Ethnicity               | 0.04     | —        |          |         |           |           |           |           |           |     |
| 3. School year             | −0.01    | −0.03    | —        |         |           |           |           |           |           |     |
| 4. First-generation status | −0.03    | 0.05     | −0.12 ** | —       |           |           |           |           |           |     |
| 5. Self-efficacy           | −0.04    | 0.06     | 0.14 *** | −0.02   | —         |           |           |           |           |     |
| 6. Perceived stress        | 0.15 *** | −0.03    | −0.06    | 0.04    | −0.41 *** | —         |           |           |           |     |
| 7. Academic satisfaction   | 0.19 *** | 0.06     | 0.12 **  | −0.07   | 0.24 ***  | −0.22 *** | —         |           |           |     |
| 8. Depressive symptoms     | −0.01    | −0.07    | −0.07    | −0.01   | −0.32 *** | 0.66 ***  | −0.27 *** | —         |           |     |
| 9. Loneliness              | −0.05    | −0.06    | −0.06    | 0.06    | −0.32 *** | 0.47 ***  | −0.18 *** | 0.57 ***  | —         |     |
| 10. Subjective health      | −0.05    | 0.03     | 0.03     | −0.01   | 0.25 ***  | −0.30 *** | 0.20 ***  | −0.28 *** | −0.31 *** | —   |
| *M*                        | 10.80    | 30.56    | 20.23    | 10.46   | 30.11     | 20.93     | 30.04     | 10.83     | 10.93     | 40.07 |
| *SD*                       | 0.40     | 10.14    | 10.10    | 0.50    | 0.46      | 0.62      | 0.69      | 0.56      | 0.63      | 0.87 |

1. Gender (1 = male, 2 = female); 2. ethnicity (1 = Asian American, 2 = Black or African American, 3 = Latina/o, 4 = Not Hispanic White, 5 = Native American, 6 = Multiracial, 7 = Other); 3. school year (1 = first year, 2 = second year, 3 = third year, 4 = fourth year); 4. first-generation status (1 = yes, 2 = no). ** $p < 0.01$. *** $p < 0.001$.

## 3. Results

### 3.1. Direct and Indirect Effects

Results of analyses showed that self-efficacy was directly related to adjustment, observed by higher academic satisfaction, decreased loneliness, and higher subjective physical health (see Figure 1). Further, self-efficacy was indirectly related to each domain of adjustment under study through lower stress (for academic satisfaction: *ab* = 0.11, 95% CI [0.06, 0.17]; for depressive symptoms: *ab* = −0.33, 95% CI [−0.39, −0.26]; for loneliness: *ab* = −0.24, 95% CI [−0.30, −0.19]; and for subjective physical health: *ab* = 0.18, 95% CI [0.11, 0.25]).

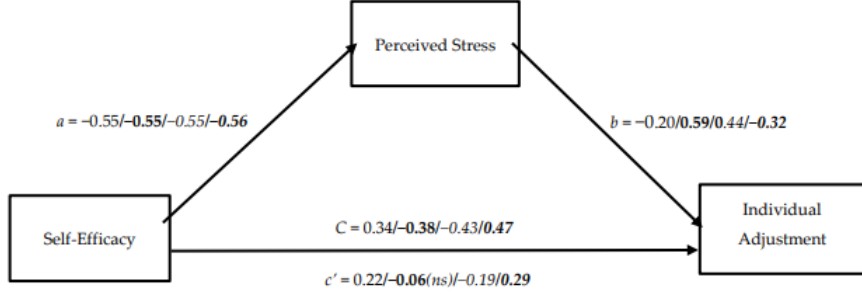

**Figure 1.** Results of the Mediator Model Testing among self-efficacy, perceived stress, and individual adjustment. Note: Four unstandardized coefficients are listed to show results for four adjustment indicators. The first is for academic satisfaction, the second (in bold) is for depressive symptoms, the third (in italics) is for loneliness, and the fourth (in italics and bold) is for subjective physical health. C = total effect of independent variable (IV) on dependent variable (DV); a = IV to mediator; b = direct effect of mediator on DV; c' = direct effect of IV on DV. All *p* < 0.001, unless noted otherwise.

*3.2. Conditional Direct and Indirect Effects*

Findings also supported the moderating roles of sex, school year, and first-generation status in the direct and indirect associations among self-efficacy, perceived stress, and adjustment. For example, associations between self-efficacy and stress (*effect* = −0.33, *t* = −2.94, *p* < 0.01, 95% CI [−0.56, −0.11]; see Figure 2) and between stress and depressive symptoms (*effect* = 0.22, *t* = 2.48, *p* = 0.013, 95% CI [0.05, 0.40]) were both stronger in women than in men. As another example, the negative association between stress and academic satisfaction was not found in the fourth-year students, although it was found in all other students (*effect* = 0.08, *t* = 1.93, *p* = 0.05, 95% CI [−0.001, 0.15]. As for first-generation students, the negative association of self-efficacy with stress was greater in first-generation college students, compared to their counterparts (*effect* = 0.24, *t* = 2.50, *p* = 0.01, 95% CI [0.05, 0.42]; see Figure 3). There was a trend for self-efficacy to positively predict academic satisfaction (this result was not apparent for their peers; effect = −0.21, *t* = −1.78, *p* = 0.07, 95% CI [−0.45, 0.02]). Race/ethnicity was not found to moderate any of the direct or indirect effects examined in the study.

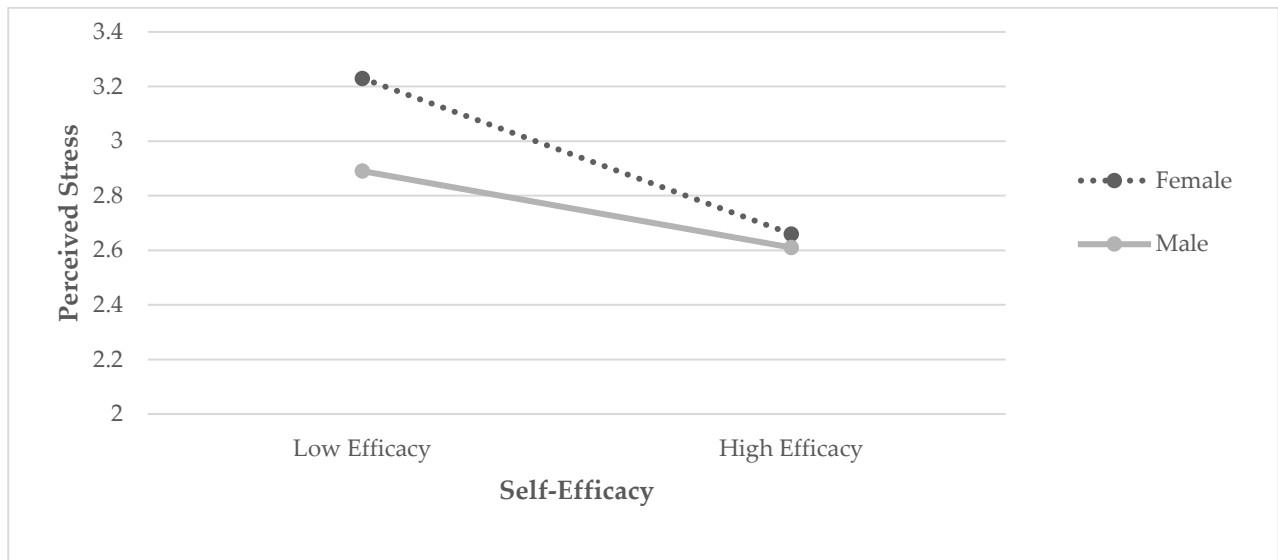

**Figure 2.** Gender moderated self-efficacy and perceived stress.

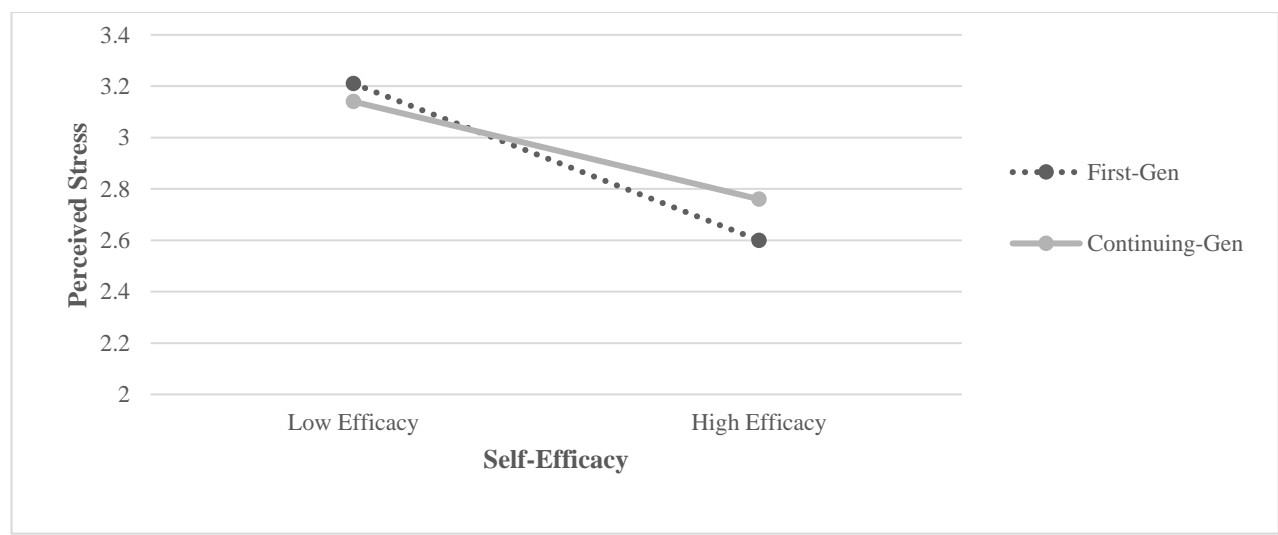

**Figure 3.** First-generation status moderated self-efficacy and perceived stress.

## 4. Discussion

Focusing on a large, ethnically diverse sample of college-attending emerging adults, we hypothesized that perceived stress would mediate the association between self-efficacy and individual adjustment. In the current study, higher self-efficacy was directly related to better adjustment observed by higher academic satisfaction [31], lower levels of loneliness [25], and higher self-ratings of physical health. The associations between self-efficacy and all adjustment indicators were indirect, i.e., through lower perceived stress.

In addition, we explored that whether the mediating association between self-efficacy, perceived stress, and individual adjustment would be moderated by sex, race/ethnicity, school year, and first-generation status. Prior research has captured how sex, race/ethnicity, school year, and first-generational college status are key variables to consider in the associations between self-efficacy, stress, and individual adjustment [4,39,41–44]. The current study showed that sex, school year, and first-generation status moderated the mediating associations between self-efficacy, stress, and individual adjustment.

In regard to sex, female students had a stronger association between self-efficacy and stress, as well as between stress and depressive symptoms, as compared to their male peers. This is consistent with prior research on the greater impact of stress on women than men [4,9,44,64]. Additionally, females in the 10- to 24-year-old age range are more likely than males to visit an emergency room for self-harm or attempt suicide [65]. Thus, understanding how to support college-attending women with handling stressors for better outcomes in adjustment is an important focus for institutions of higher education. Colleges can consider building in content on handling stressors to orientation programs or new student seminars, as well as collaborating with sex-specific resources such as women's centers, sororities, and female sports teams on campus. In particular, a number of recent high-profile suicides of female college student athletes has brought attention to the challenges faced by this subpopulation [66]. For those institutions with single-sex residence halls or floors, this could be a topic of focus for resident assistants and residential life staff.

School year is an aspect that is not often discussed, and when it is, the focus is generally on first-year or last-year students. In the current study, there was evidence that for first-, second-, and third-year students, there was a negative association between perceived stress and academic satisfaction. However, perceived stress was not correlated with academic satisfaction among the fourth-year students. It could be that younger students take time to get adjusted to the college environment while encountering stressor events, whereas seniors are more experienced at dealing with stress; they have been able to adjust to it before and remain enrolled in their studies. Perceived stress indicates that an individual feels taxed by the environment, and more experienced students may feel an enhanced confidence in their capabilities. Both age and experience may also help to build self-efficacy, which may assist in reducing the expected negative impact of stress on academic satisfaction. Alternatively, seniors could be "looking ahead" to their graduation date and life after graduation; perhaps seniors' experiences with stress would be associated with other aspects of their lives not measured in the present study.

About a third of all college students in the United States are first-generation [67]. The current study provides initial evidence that first-generation status moderated the negative association of self-efficacy and stress, with it being greater for first-generation college students compared to their peers. In addition, self-efficacy was positively related to academic satisfaction for first-generation students, whereas no relation was found for continuing-generation students. As its negative correlation with stress levels and positive correlation with satisfaction and school performance was found in our study, self-efficacy has been shown to be incredibly important for first-generation students [31], for whom having confidence and self-belief may be particularly important if they do not have family to help them navigate the college environment. Clearly, colleges should consider first-generation specific programs to help students build self-efficacy. One way of doing this might be through a mentoring approach, pairing first-year first-generation students with their higher-year first-generation peers to model success, creating an environment that

encourages growth over perfection, and fostering goal-setting and self-reflection both within and outside the classroom.

*4.1. Limitations*

Although our sample was relatively large and ethnically diverse, the participants primarily identified as female (80%). Thus, data on an increased number of male students would have been helpful for comparison, particularly given that male students are less likely to both pursue and complete higher education than their female peers [68]. Further, the current study was conducted at a large, public university in the Northeastern United States with a largely in-state population. Generalizations may not be applicable to students in different regions or countries, or to students from private institutions.

It is also important to note that this study utilized cross-sectional data, and inferences about causality are not warranted. Future research on these topics should consider a longitudinal design. Finally, this study could be replicated with a sample of emerging adults that are not attending college for comparison. Non-college-attending emerging adults are an understudied population [69], and a better understanding of their experiences with self-efficacy, stress, and their work towards meeting occupational goals would be valuable.

Despite these limitations, the current study provides key insights on the associations among self-efficacy, perceived stress, and individual adjustment. This paper is among the first to examine these issues together, with an additional focus on first-generation status, sex, race, and academic year, in a diverse sample of college-attending emerging adults. Stress and adjustment are important issues in higher education today, particularly in the face of the COVID-19 pandemic. Having a further understanding of the association between stress and individual adjustment, as well as the buffering effects of self-efficacy, for different populations of students can help educators and administrators prepare to meet these challenges and know how best to support their students. Extending the findings of this research would build on the literature, which would inform practical supports for college students as they navigate emerging adulthood.

*4.2. Implications for Practice*

The current findings suggest that self-efficacy is impactful for the adjustment of college-attending emerging adults. Prior to college, parents and educators should collaborate to promote practical skills and self-confidence, as these factors have been shown to predict self-efficacy [70,71]. On campus, efforts can be made to help decrease students' stress, reinforce their self-concept, and make them aware of campus-based resources that can support them. For example, making resources like college counseling and peer mentors visible and easily accessible would be an asset in facilitating success and positive adjustment in (potentially) stressed students. Additionally, higher education administrators and educators should consider ways to build self-efficacy in the first year, to better serve students throughout their college career. Orientation programs and first-year seminars, as well as residential life staff for first-year students, can include this topic in their sessions, workshops, and curricula. The current results show that self-efficacy is vital to the success of first-generation students, and institutions should pay special attention to this group; for instance, colleges may want to create summer bridge or peer mentoring programs to specifically support first-generation students with the transition to the college environment, as well as to bolster their self-efficacy.

Although retention is important for first-year students, as about 30% of college students withdraw enrollment after the first year [72], retention throughout all four years is an important consideration for practitioners, given the overall rate of 56% of college students who begin their studies and do not complete their degree [72]. As noted previously, the transition from high school to college is a significant challenge [4,25,31,36,39,48]. However, given our findings that perceived stress is negatively related to academic satisfaction for first-, second-, and third-year students, sustained enhanced retention efforts throughout the first three years of college seem warranted. Thus, peer-counseling or mentoring programs

led by college seniors could potentially be an important component of support for their less experienced peers.

## 5. Conclusions

Results from this study suggest that higher self-efficacy is related to better adjustment for college-attending emerging adults. This finding was especially salient for female and first-generation students in the current sample. Given the role that both self-efficacy and perceived stress play in the adjustment of college students, the current study lends more evidence to support the need of parents, K-12 educators, and higher education officials to help foster self-efficacy in youth and equip them to manage stress. In particular, the need to support women, particularly around perceived stress, is an important consideration given their increased rates of college attendance. Finally, first-generation students are a critical population, as prior research has shown that they are less likely to complete college [7,52]. Thus, a better understanding of the roles of self-efficacy, stress, and adjustment for this population will allow colleges to tailor their supports for this group.

**Author Contributions:** Conceptualization, R.C.M., P.B.P., S.E.G. and C.-Y.S.L.; methodology, C.-Y.S.L.; validation, C.-Y.S.L.; formal analysis, C.-Y.S.L.; investigation, C.-Y.S.L.; resources, C.-Y.S.L.; data curation, C.-Y.S.L.; writing—original draft preparation, R.C.M., P.B.P., S.E.G. and C.-Y.S.L.; writing—review and editing, R.C.M., P.B.P., S.E.G. and C.-Y.S.L.; visualization, R.C.M. and C.-Y.S.L.; supervision, C.-Y.S.L.; project administration, R.C.M. and C.-Y.S.L. All authors have read and agreed to the published version of the manuscript.

**Funding:** This research received no external funding.

**Institutional Review Board Statement:** The study was conducted in accordance with the Declaration of Helsinki and approved by the Institutional Review Board of MONTCLAIR STATE UNIVERSITY for studies involving humans.

**Informed Consent Statement:** Informed consent was obtained from all subjects involved in the study.

**Data Availability Statement:** Our study is not based on public use data. For access to the data used in the study, please contact the 4th author.

**Conflicts of Interest:** The authors declare no conflict of interest.

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
