# Peer review of "Self-Efficacy, Perceived Stress, and Individual Adjustment among College-Attending Emerging Adults"

_2673-995X, doi:10.3390/youth2040047_

Round 1
Reviewer 1 Report
This manuscript is fluently written, logically rigorous, and detailed in data analysis. I think the manuscript is worthy of publication, but there are several issues that need to be addressed before publication.
1. There is a lot of research on this topic in general, so Identify what makes this article different from the rest of studies that are available in the literature. Identify the gap in exiting literature, by arguing what is missing or inadequate in existing solutions and thus your study is necessary.
2. Before developing the conceptual framework of the current research, the paper should first engage with existing theoretical frameworks in literature to demonstrate the need for the proposed framework. What literature is there to support this claim? This aspect of the paper is one of the weakest and needs in-depth relook to strengthen the theoretical aspect.
3. The authors need to add some comparisons of the results of related studies in other countries.
Reviewer 2 Report
The current study examines the associations between self-efficacy and individual adjustment directly and indirectly through perceived stress. At the same time, it analyzes moderate mediation defects by gender, ethnicity, and school year.
The results are of great interest and the introductory approach and methodological design is correct.
The conclusions respond to the objectives.
Author Response
Thank you for your comments. We have made some minor revisions based on another reviewer's comments and will be submitting an updated manuscript for your review.